# Targeting Mitochondrial Metabolism as a Strategy to Treat Senescence

**DOI:** 10.3390/cells10113003

**Published:** 2021-11-03

**Authors:** Yun Haeng Lee, Ji Yun Park, Haneur Lee, Eun Seon Song, Myeong Uk Kuk, Junghyun Joo, Sekyung Oh, Hyung Wook Kwon, Joon Tae Park, Sang Chul Park

**Affiliations:** 1Division of Life Sciences, College of Life Sciences and Bioengineering, Incheon National University, Incheon 22012, Korea; licdldbsgod@naver.com (Y.H.L.); ww1304@naver.com (J.Y.P.); 523042@naver.com (H.L.); sos002645@naver.com (E.S.S.); lbl646@gmail.com (M.U.K.); kingdom0304@naver.com (J.J.); 2Department of Medical Sciences, Catholic Kwandong University College of Medicine, Incheon 22711, Korea; ohskjhmi@gmail.com; 3The Future Life & Society Research Center, Chonnam National University, Gwangju 61186, Korea

**Keywords:** mitochondrial metabolic reprogramming, mitochondria, ROS, senescence amelioration

## Abstract

Mitochondria are one of organelles that undergo significant changes associated with senescence. An increase in mitochondrial size is observed in senescent cells, and this increase is ascribed to the accumulation of dysfunctional mitochondria that generate excessive reactive oxygen species (ROS). Such dysfunctional mitochondria are prime targets for ROS-induced damage, which leads to the deterioration of oxidative phosphorylation and increased dependence on glycolysis as an energy source. Based on findings indicating that senescent cells exhibit mitochondrial metabolic alterations, a strategy to induce mitochondrial metabolic reprogramming has been proposed to treat aging and age-related diseases. In this review, we discuss senescence-related mitochondrial changes and consequent mitochondrial metabolic alterations. We assess the significance of mitochondrial metabolic reprogramming for senescence regulation and propose the appropriate control of mitochondrial metabolism to ameliorate senescence. Learning how to regulate mitochondrial metabolism will provide knowledge for the control of aging and age-related pathologies. Further research focusing on mitochondrial metabolic reprogramming will be an important guide for the development of anti-aging therapies, and will provide novel strategies for anti-aging interventions.

## 1. Introduction

Senescence is characterized by a condition in which somatic cells lose their capacity to proliferate after a limited number of mitotic divisions [1]. Moreover, senescence is defined as changes in the shape and function of organelles, the most prominent of which occur in mitochondria. Specifically, mitochondria undergo structural changes associated with significant increases in size and volume [2]. Such increases are due to the accumulation of mitochondria that produce reactive oxygen species (ROS) as a by-product of inefficient electron transport in the electron transport complex (ETC) [3]. The energy transferred by flowing electrons transports protons across the inner mitochondrial membrane (IMM), which in turn forms an electrochemical proton gradient that is used to make adenosine triphosphate (ATP) during oxidative phosphorylation (OXPHOS). Dysfunctional mitochondria are not only a major producer of excessive ROS, but also a major target of ROS-induced ETC damage, which impairs OXPHOS efficiency [4]. Thus, changes in mitochondrial metabolism occur during senescence, demonstrating that senescent fibroblasts rely less on OXPHOS but more on glycolysis as a source of energy [5]. This finding is supported by observations that alterations in mitochondrial metabolism contribute to premature reductions in organ function [6]; however, the underlying mechanisms for reprogramming altered mitochondrial metabolism remain elusive. Therefore, basic knowledge of mitochondrial metabolic reprogramming and mechanism-based strategies for regulating mitochondrial metabolism are needed.

This review aims to discuss changes in mitochondrial metabolism during senescence, and suggest a role for mitochondrial metabolic reprogramming in the regulation of senescence. An extensive literature search was performed on PubMed (a search engine that accesses the MEDLINE database) using search terms such as senescence-related mitochondrial dysfunction, senescence-related metabolic alteration, and mitochondrial metabolic reprogramming. A systematic review strategy, which is the gold standard in medical review writing, was employed to analyze the retrieved results [7]. Based on previous and recent findings, we herein provide new insights into the crosstalk between mitochondrial metabolism and senescence, and propose mitochondrial metabolic reprogramming as a therapeutic target for aging and age-related diseases.

## 2. Mitochondrial Alterations Associated with Senescence

Mitochondria are organelles that undergo a continuous cycle of fusion and division, called “mitochondrial dynamics”, to maintain proper function [8]. Mitochondrial fusion consists of two steps: the outer mitochondrial membrane (OMM) is fused by mitofusin 1 (Mfn1) and mitofusin 2 (Mfn2), and then the IMM is fused by mitochondrial dynamin-like GTPase (OPA1). Mitochondrial fusion mixes the contents of partially damaged and healthy mitochondria, contributing to a relatively homogeneous network (Figure 1A). This process helps in quality control by maintaining mitochondrial integrity and homeostasis, especially under environmental and metabolic stress. Mitochondrial fission is initiated by membrane contraction by the endoplasmic reticulum (ER), wherein receptors on OMM, including fission 1 protein (FIS1), mitochondrial fission factor (Mff), and mitochondrial dynamics proteins of 49 kDa and 51 kDa (MiD49 and MiD51, respectively), recruit the fission mediator, dynamin-related protein 1 (Drp1), to the mitochondrial surface [9,10,11,12]. Drp1 assembles around the mitochondrial surface to form higher-order oligomers, which divide the mitochondrion into two mitochondria [13]. Thus, mitochondrial fission produces new mitochondria, providing a sufficient number of them for cell growth and division (Figure 1A). Mff–Drp1 interaction is important for mitochondrial fission, as evidenced by the discovery that inhibitors of the Mff–Drp1 interaction interfere with fission, leading to mitochondrial dysfunction [14]. However, the Mff–Drp1 interaction does not prevent Drp1 from interacting with other receptors [14]. A recent study demonstrated a novel role for FIS1 in mitochondrial fission by showing that FIS1 stimulates mitochondrial fission by preventing mitochondrial fusion through the inhibition of GTPase activity in Mfn1, Mfn2 and OPA1 [15]. Mitochondrial fission is also a form of quality control, in which defective mitochondria are removed through autophagy, preventing the accumulation of defective mitochondria [16]. Thus, a fine-tuned balance between fusion and fission allows for the maintenance of mitochondrial quality [17].

Alteration in mitochondrial morphology is a pathogenic hallmark of senescence (Table 1). Defects in lysosomal function due to senescence prevent lysosomal enzymes from targeting autophagosomes, leading to defects in the removal of dysfunctional mitochondria [18,19]. Thus, dysfunctional mitochondria are not eliminated efficiently, but accumulate [18,19]. Dysfunctional mitochondria are vulnerable to further oxidative damage and constitute a major source of excessive ROS. Increased oxidative damage depletes the mitochondrial fission regulator, FIS1, and disrupts the mitochondrial fusion/fission balance, resulting in the formation of giant mitochondria featuring highly interconnected networks [20] (Figure 1B). The enlarged mitochondria limit the effectiveness of quality control, in which damaged mitochondria are efficiently eliminated through autophagy [20,21]. Extending the relevance of these findings, a significant increase in the proportion of giant mitochondria was observed in the livers of aged mice, with an average increase in mitochondrial size of 60% [22]. Increases in mitochondrial size are also induced by a feedback mechanism to compensate for reduced mitochondrial function due to ROS damage, as excessive ROS deteriorates ETC function and subsequently dissipates mitochondrial membrane potential (ΔΨm) [23]. This phenomenon is reinforced by findings showing that large aggregates of mitochondria with low ΔΨm and impaired ATP production are frequently observed in senescent endothelial cells (ECs) [24].

The accumulation of mitochondria with various defects correlates strongly with the pathogenesis of aging and age-related diseases. For example, the excessive ROS generated by dysfunctional mitochondria activates the p53 and pRb pathways, resulting in permanent cell cycle arrest and the exacerbation of cellular senescence [41,42,43,44]. Excessive ROS also activates polyADP-ribose polymerase 1, a NAD^+^ consuming enzyme [45]. This leads to a significant reduction in the NAD^+^ levels and NAD^+^/NADH ratio in parallel with increased oxidative stress and decreased antioxidant capacity, consequently exacerbating senescence [46]. Furthermore, ROS can damage the proteins involved in mitochondrial proteostasis, which play an important role in maintaining and regulating protein quality within mitochondria [47]. Defects in mitochondrial proteostasis impair protein folding and the processing of misfolded or aggregated proteins, leading to aging and age-related diseases through accelerated proteostatic collapse [47,48]. The causal relationship between senescence and dysfunctional mitochondria is supported by findings that the accumulation of dysfunctional mitochondria leads to detrimental effects, including altered cellular homeostasis and degenerative changes in tissues [49,50,51,52]. Support for the causal relationship is evident in the finding that extensive targeted mitochondrial depletion inhibits ROS generation and the secretion of key senescence-associated secretory phenotype (SASP) factors, such as IL-6 and IL-8 [53]. Mitochondria are required for the development of the pro-oxidant and pro-inflammatory features of senescence, suggesting that mitochondria are candidate therapeutic targets for reducing the deleterious effects of senescence [53]. Taken together, these results suggest that dysfunctional mitochondria are not only a concomitant phenomenon of senescence, but also a cause of senescence, implying the existence of a vicious cycle involving multiple feedback loops rather than a linear causal relationship [54].

Mitochondria also become targets of toxin-induced ROS, which damage mitochondria and impair mitochondrial function (Table 1). For example, rotenone is a mitochondrial complex I inhibitor that interrupts electron transfer from the iron–sulfur clusters of complex I to coenzyme Q10 (CoQ) [55]. Once inside the mitochondria, rotenone acquires electrons from complex I for the redox cycle, forming excess mitochondrial ROS. The impairment of complex I by rotenone induces senescence-related Parkinson’s disease [56]. Another example is antimycin A, a mitochondrial complex III inhibitor that inhibits electron transfer from cytochrome b to cytochrome c1 in ETC [57]. In this process, antimycin A leaks a single electron to O_2_ to generate mitochondrial ROS through a non-enzymatic reaction. The oxidative stress induced by antimycin A deteriorates mitochondrial function and induces premature senescence [25].

## 3. Alterations in Mitochondrial Metabolism Associated with Senescence

Mitochondria generate the chemical energy needed to power biochemical reactions in the form of ATP, and the amount of ATP accounts for about 90% of ATP production in the cell [58]. ATP production in mitochondria is accomplished by a sequential reaction called OXPHOS, which involves four ETC complexes (complexes I through IV) and ATP synthase [59]. During OXPHOS, the redox process of the ETC produces a hydrogen ion (H^+^) concentration gradient, leading to the movement of H^+^ from ATP synthase to the matrix to generate ATP (Figure 2A).

Alterations in OXPHOS function are observed in various models of senescence [30,31] (Table 1). As senescence progresses, dysfunctional mitochondria produce excess ROS, causing the unwanted oxidation of proteins involved in OXPHOS and impairment of their function [60] (Figure 2B). Thus, electron transport in the ETC is disturbed and electrons leak out of the ETC. Inefficient electron transport concurrently impairs proton transport through the IMM and dissipates ΔΨm, thereby reducing the efficiency of OXPHOS and accompanying a lack of ATP production (Figure 2B). The leaked electrons also react with O_2_, generating excessive mitochondrial ROS [61]. The deterioration of OXPHOS function with senescence is evidenced by findings that senescence induces the deterioration of the ETC complexes in liver, brain, and muscle tissues, leading to a decrease in mitochondrial respiratory function [26,27,28].

Alterations in OXPHOS lead to the development of aging and age-related diseases, which suggests a causal relationship between senescence and OXPHOS deterioration (Table 1). For example, a mouse model of senescence produced by *mev-1* (ortholog of the complex II) mutation exhibits deterioration of OXPHOS accompanying precocious age-dependent corneal physiological changes [29]. Support for this phenomenon is evident from observations that iron chelation with deferoxamine reduces complex II activity through the translational inhibition of iron–sulfur clusters in complex II [30,31]. Decreases in complex II activity sustain the disruption of ΔΨm with significantly reduced intracellular ATP levels prior to the acquisition of the senescence phenotype. Similarly, the inhibition of complex IV activity by transforming growth factor β1 (*TGF-β1*) induces mitochondrial ROS generation and the persistent disruption of ΔΨm. Thus, the *TGF-β1*-mediated inhibition of complex IV directly triggers senescence arrest in mink lung epithelial cells through prolonged mitochondrial ROS generation and decreased ATP generation, confirming the causal relationship between senescence and OXPHOS deterioration [32,33,34].

Alterations in mitochondrial metabolism with decreasing dependence on OXPHOS but increasing dependence on glycolysis constitute one of the characteristic changes observed with senescence [62,63] (Table 1). The higher reliance on glycolysis during replicative senescence is attributed to the less energetic state reflected by the marked decrease in ATP levels in senescent cells [37]. Specifically, glycolysis is upregulated to generate additional ATP to compensate for the loss of energy production in dysfunctional mitochondria [35]. Furthermore, the analysis of energy metabolism in senescent cells reveals an increase in glucose consumption and lactic acid production, indicating that glycolysis is actively proceeding [36]. This phenomenon is further assessed by metabolic profiling, which reveals age-dependent changes in mitochondrial metabolism, marked by significant transitions to more glycolytic states [37].

Mitochondrial Ca^2+^ homeostasis plays an important role in the regulation of mitochondrial metabolism [23,64] (Table 1). This homeostasis is regulated by protein channels localized in the IMM and OMM, and also by crosstalk with the ER [65] (Figure 3A). Mitochondrial Ca^2+^ influx occurs through porin-like proteins called voltage-dependent anion channels (VDAC) in the OMM. Then, Ca^2+^ enters the mitochondrial matrix through the mitochondrial calcium uniporter (MCU) in the IMM. The mitochondrial Ca^2+^ efflux from the mitochondrial matrix is driven by two channels, a H^+^/Ca^2+^ exchanger (HCX) and a Na^+^/Ca^2+^ exchanger (NCLX), present in the IMM. Intracellular Ca^2+^ buffering from the ER to mitochondria is achieved by inositol 1,4,5-trisphosphate receptor (IP_3_R)–Grp75–VDAC interaction [66,67,68] (Figure 3A). Grp75, linking IP_3_R in the ER with VDAC in the mitochondria, creates a tight juxtaposition between the ER and mitochondria to regulate intracellular Ca^2+^ buffering [66,67,68]. During senescence, IP_3_R in the ER and VDAC1/MUC in the mitochondria act as senescence regulators by controlling the concentration of mitochondrial Ca^2+^. In particular, senescence triggers IP_3_R to release Ca^2+^ from the ER and causes VDAC/MCU channels to initiate the inward flow of Ca^2+^, leading to mitochondrial Ca^2+^ overload [38] (Figure 3B). Mitochondria overloading with Ca^2+^ causes the collapse of the electron transport in the ETC, resulting in increased electron leak and consequent mitochondrial ROS generation [38]. An increase in mitochondrial ROS due to mitochondrial Ca^2+^ overload induces the sustained opening of the mitochondrial transition pore (mPTP) [39]. Then, mPTP opening causes a rapid collapse in ΔΨm and swelling of the mitochondria, resulting in the loss of cytochrome c, a component of the ETC that transports electrons to complex IV. Inefficient electron transport by leaked electrons results in a deficiency in the generation of electrochemical proton gradient, leading to decreased OXPHOS efficiency [40] (Figure 3B). Furthermore, the frequency and duration of mPTP opening increases with the progression of senescence, and increased mPTP activity is associated with several neurodegenerative diseases [23]. Mitochondrial Ca^2+^ overload appears to be the initiator of alteration in mitochondrial metabolism, which contributes to the deficits observed during senescence and neurodegeneration [23]. Thus, the maintenance of an appropriate level of mitochondrial Ca^2+^ concentration to regulate mitochondrial metabolism might be a novel strategy for the treatment of senescence.

## 4. Targeting Mitochondrial Metabolism as a Strategy to Treat Senescence

As mentioned in Section 3, senescence triggers mitochondrial metabolic alteration from OXPHOS to glycolysis, and senescent cells exhibit greater dependence on glycolysis as an energy source [69]. The close relationship between mitochondrial metabolism and senescence is demonstrated by the discovery that alterations in mitochondrial metabolism provoke premature reductions in tissue and organ function [6], whereas improvements in OXPHOS efficiency extend the lifespan of cells and organisms [70]. These interconnections suggest that regulatory mechanisms of mitochondrial metabolism are essential for adequately controlling senescence [71,72]. Therefore, herein, we systematically characterize and propose a potential therapeutic strategy targeting mitochondrial metabolism to induce mitochondrial metabolic reprogramming for the treatment of senescence.

Activation of OXPHOS coupled with increased ATP production supports the importance of mitochondrial metabolic reprogramming in regulating senescence. Specifically, senescent cells exhibit a deficiency in CoQ, which accepts electrons from complex I/II and transfers them to complex III in the ETC (Figure 4A; green CoQ indicates a deficiency in CoQ) [73]. Thus, electron transport in ETC is perturbed in cells with CoQ deficiency, resulting in electron leakage with loss of ΔΨm [73]. Electrons that leak from the ETC prematurely react with O_2_, causing excessive mitochondrial ROS production [61] (Figure 4A). In agreement with this finding, CoQ activates a proton-motive Q cycle, allowing complex III to pump protons from the mitochondrial matrix into the intermembrane space, generating a proton motive force ΔΨm [74]. A deficiency of CoQ reduces the efficiency of OXPHOS based on ΔΨm in flowing protons back to the mitochondrial matrix via ATP synthase, resulting in decreased ATP production [74]. Given that CoQ plays an important role in mitochondrial OXPHOS and ATP production, CoQ-deficient fibroblasts were treated with CoQ [75]. CoQ treatment improves mitochondrial metabolism, which manifests as a significant increase in ATP production and a significant decrease in mitochondrial ROS generation [75] (Figure 4B; pink CoQ indicates a higher level of CoQ in the IMM). The beneficial effects of CoQ supplementation on OXPHOS are further supported by animal model experiments using senescence-accelerated mice [76]. CoQ supplementation improves OXPHOS efficiency by increasing complex I/IV activity and subsequently decreasing mitochondrial ROS generation. Additionally, CoQ supplementation slows the progression of aging-related symptoms and prevents aging, suggesting that strategies to activate OXPHOS efficiency may be effective in treating senescence [76].

There are other strategies that modulate OXPHOS efficiency to induce mitochondrial metabolic reprogramming for senescence amelioration. Long-term caloric restriction (CR) increases complex IV levels, resulting in the partial compensation of electron leakage, thereby decreasing ROS generation [77]. A CR mimetic, epigallocatechin 3-gallate (EGCG), rescues the catalytic activity of complex I/ATP synthetase and restores OXPHOS efficiency [78]. In addition, EGCG acts as an activator of sirtuin 1 (SIRT1), a protein deacetylase, and reduces the acetylation of PGC-1α. Thereby, EGCG activates PGC-1α, which regulates mitochondrial biosynthesis and function [78]. Consistent with this finding, resveratrol (RSV) induces SIRT1-dependent PGC-1α deacetylation [79]. RSV significantly increases OXPHOS efficiency, as evidenced by the induction of OXPHOS genes, including the ETC complexes, ATP synthase, and respiratory apparatus [80]. Furthermore, RSV prolongs lifespan by enabling the long-term supply of ATP through the restoration of metabolic homeostasis [80]. In line with this finding, dietary supplementation containing essential and branched-chain amino acids (called PD-0E7) deacetylates and activates PGC-1α to amplify mitochondrial responses. Accordingly, PD-0E7 upregulates the activity of complexes I, II, and IV, and improves muscular and cognitive performance in the senescence-accelerated mouse prone 8 model [81]. Finally, boosting NAD^+^ levels with nicotinamide riboside (NR) constitutes an efficient way to increase OXPHOS efficiency. NR supplementation increases NAD^+^ levels and the NAD^+^/NADH ratio, which are known to be significantly reduced in senescent cells [82]. Increased NAD^+^ levels by NR stimulate mitophagy to remove damaged mitochondria and enhance OXPHOS efficiency by upregulating basal/maximal ATP-linked oxygen consumption rates [83]. Concurrently, improved mitochondrial function by NR prevents senescence and SASP [83].

Targeting a higher glycolytic state of senescent cells is an alternative strategy that highlights the importance of mitochondrial metabolic reprogramming in regulating senescence. *PFKFB3* is a gene that encodes 6-phosphofructo-2-kinase/fructose-2,6-biphosphatase 3, which functions as a pivotal activator of glycolysis by activating phosphofructokinase 1 (PFK1), converting fructose-6-phosphate to fructose-1,6-bisphophate [84]. The pharmacological inhibition of the glycolytic activator, PFKFB3, inactivates glycolysis and alleviates age-related cerebral ischema/reperfusion injury in mice [85] (Figure 4B). The effectiveness of the strategy to limit glycolysis upon senescence amelioration is further supported by experiments using D-glucosamine (GlcN), which inhibits the activity of glyceraldehyde-3-phosphate-dehydrogenase (GAPDH) in the glycolytic pathway. GlcN increases mitochondrial respiration by promoting the dependence of energy metabolism on OXPHOS, while impairing glycolysis, thus prolonging lifespan in many species, including mammals [86] (Figure 4B). While inhibiting the specific enzymes required for glycolysis may be a strategy to control senescence, there is growing evidence that inhibiting the entire glycolysis process may serve as a platform to regulate senescence. For example, the impaired glucose metabolism, induced by glucose restriction, improves mitochondrial OXPHOS and consequently prolongs the lifespan of *Caenorhabditis elegans* [87].

Mitochondria play an important role in regulating intracellular metabolism, serving as a platform for receiving signals from key elements of cells and tissues [72]. Several cellular signaling pathways are directly or indirectly linked with mitochondrial metabolism [88]. Targeting such pathways might be an alternative strategy to induce mitochondrial metabolic reprogramming. These pathways involve ataxia telangiectasia mutated (ATM), rho-associated protein kinase (ROCK), and serine/threonine protein kinase B-Raf (BRAF) (Figure 4C). Targeting these pathways has been evaluated to be effective in regulating senescence through the modulation of the mitochondrial metabolism.

(i)Targeting ATM signal pathway.

ATM controls lysosomal pH by regulating the assembly/disassembly of the V_1_ and V_0_ domains in the V-ATPase proton pumps present in lysosomal membranes [62]. Notably, the inhibition of the ATM signaling pathway promotes V_1_-V_0_ assembly, leading to the re-acidification of lysosomes. In turn, this leads to a functional restoration of the lysosome, thus enhancing the clearance of dysfunctional mitochondria, a key mechanism that maintains mitochondrial function [89]. The restoration of mitochondrial function by ATM inhibition is accompanied by mitochondrial metabolic reprogramming from glycolysis to OXPHOS, resulting in increased ATP production and the restoration of senescence-related phenotypes [62,90]. The effect of targeting the ATM signaling pathway, indirectly linked to mitochondrial metabolism, highlights the importance of mitochondrial metabolic reprogramming in the regulation of senescence, and may have clinical applications in controlling aging and age-related diseases.

(ii)Targeting ROCK signal pathway.

ROCK controls mitochondrial ROS generation while regulating the interaction between Rac1b and cytochrome c [63]. The inhibition of the ROCK signaling pathway reduces mitochondrial ROS production and simultaneously interrupts electron transfer from cytochrome c, preventing the partial reduction of O_2_ [63]. Reducing oxidative damage by modulating ROCK activity enhances complex IV activity in the ETC, leading to improved mitochondrial function along with metabolic reprogramming [63,91]. The importance of metabolic reprogramming via the regulation of ROCK activity is elucidated by experiments using metabolic reprogrammers that artificially shift the metabolism from OXPHOS to glycolysis. Artificial metabolic reprogramming to glycolysis interferes with ROCK inhibition-mediated senescence improvement, suggesting that metabolic reprogramming by modulating ROCK activity plays a prerequisite role in senescence regulation [63,91]. The effect of targeting the ROCK signaling pathway, directly linked to mitochondrial metabolism, also supports the significance of mitochondrial metabolic reprogramming as a therapeutic strategy for treating senescence.

(iii)Targeting BRAF signal pathway.

BRAF regulates MAP kinase/ERK signaling pathways and modulates the activity of substrates in response to various signals to maintain cellular homeostasis [92]. The inhibition of the BRAF signaling pathway increases mitophagy, which regulates mitochondrial quality by eliminating dysfunctional mitochondria [93]. Concurrently, ROS generation is decreased by modulating BRAF activity, which leads to mitochondrial function restoration with an increase in OXPHOS and a decrease in glycolysis. Furthermore, metabolic reprogramming induced by the inhibition of BRAF activity is a prerequisite for ameliorating senescence, as evidenced by the discovery that artificial metabolic reprogrammers block senescence amelioration induced by BRAF inhibition.

The maintenance of mitochondrial metabolism through mitochondrial Ca^2+^ homeostasis represents a therapeutic strategy to induce mitochondrial metabolic reprogramming for senescence amelioration. Adequate levels of mitochondrial Ca^2+^ activate the enzymatic activity of the ETC and stimulate the entire OXPHOS machinery [94], whereas mitochondrial Ca^2+^ overload generates excessive ROS and induces metabolic derangement [23]. As the fine-tuning of mitochondrial Ca^2+^ concentration initiates mitochondrial metabolic reprogramming [95], maintaining adequate levels of mitochondrial Ca^2+^ can be a novel treatment for senescence. For example, myocardial reperfusion is an age-related disease that manifests as decreased resistance to myocardial reperfusion injury in the elderly [96]. Myocardial reperfusion is also associated with alterations in mitochondrial Ca^2+^ homeostasis and mitochondrial metabolism [97]. As the transport of Ca^2+^ from the cytoplasm to the mitochondria is facilitated by MCU in the IMM, mitochondria from the reperfused hearts are treated with the cell-permeable MCU inhibitor, Ruthenium 360 (Ru360) [98,99] (Figure 4D). The inhibition of MCU by Ru360 improves OXPHOS efficiency by maintaining mitochondrial Ca^2+^ at basal levels and reducing the proportion of mitochondria exhibiting Ca^2+^ overload. Improving mitochondrial metabolism is accompanied by the restoration of the pathophysiological symptoms of the reperfused heart to a physiological state [99]. The importance of maintaining adequate levels of mitochondrial Ca^2+^ homeostasis is further supported by the discovery that the inhibition of MCU by microRNA-mediated silencing protects cardiomyocytes from oxidative damage and restores mitochondrial function [100] (Figure 4D).

## 5. Conclusions and Perspectives

The role of mitochondrial metabolic reprogramming in senescence has been analyzed and reviewed by several researchers. One review specifically presented senescence-related mitochondrial metabolic changes and their effects on the modulation of the immune response underlying senescence [101]. Although the importance of mitochondrial metabolism in senescence has been discussed, it has been limited to its role in ECs. Cell-specific limitations were addressed in another review dealing with the regulation of mitochondrial metabolism in several aging models [102]. This review focused on the mechanistic involvement of metabolic regulators in several aging models, and has proposed metabolic switches as modulators of senescence. The causes and consequences of mitochondrial metabolic changes have been detailed, but no specific strategies to induce metabolic reprogramming have been proposed. Recently, a review paper highlighted cellular metabolism as one of the regulatory factors controlling various senescence-related phenotypes [103]. This review focused on the associations between metabolism and senescence-related phenotypes. A putative mechanism that can target metabolic differences between young and senescent cells has been proposed as a strategy to eliminate the deleterious effects of senescent cells. However, considering that many cellular pathways are damaged by senescence, it is not clear whether therapeutic approaches that restore only mitochondrial metabolism will be effective in the treatment of senescence.

In this review, we discussed and summarized senescence-related mitochondrial dysfunction and consequent mitochondrial metabolic alterations. We further assessed the causal relationship between mitochondrial metabolism and senescence, and suggested several therapeutic strategies targeting mitochondrial metabolism for the treatment of senescence. Changes in mitochondrial metabolism occur during senescence, and restoring mitochondrial metabolism to a normal state by several approaches restores senescence and senescence-associated phenotypes. Furthermore, mitochondrial metabolic reprogramming based on multiple approaches serves as a prerequisite for aging treatment. Thus, the appropriate regulation of mitochondrial metabolism, which does not rely on one approach, may open up the possibility of addressing the pathological symptoms of senescence, by which many cellular pathways are damaged. Here, we propose mitochondrial metabolic reprogramming as a promising therapeutic target for the treatment of senescence. Further studies of the molecular mechanisms that support the role of mitochondrial metabolic reprogramming in the initiation and progression of senescence will provide novel therapeutic strategies for aging and age-related diseases.

## Figures and Tables

**Figure 1 cells-10-03003-f001:**
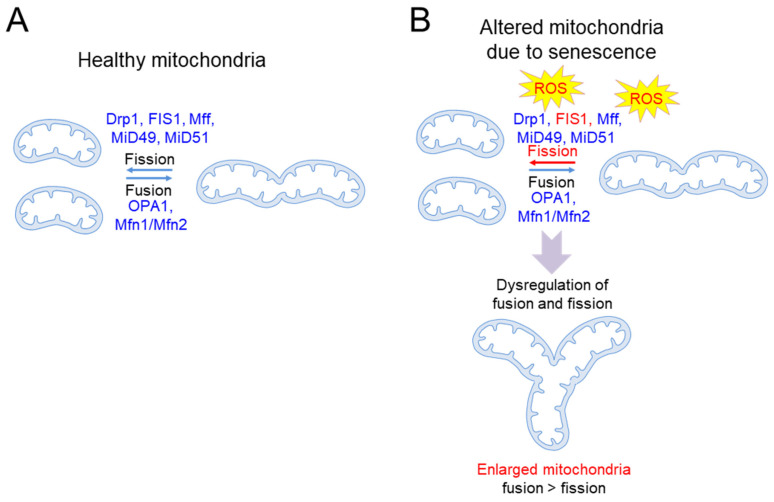
Schematic representation of basic mechanisms of senescence-induced mitochondrial damage. (**A**) Mitochondria are organelles that undergo a continuous cycle of fusion and division. The proteins involved in mitochondrial fusion include mitofusin 1 (Mfn1), mitofusin 2 (Mfn2), and mitochondrial dynamin-like GTPase (OPA1). Proteins involved in mitochondrial fission include fission 1 protein (FIS1), mitochondrial fission factor (Mff), mitochondrial dynamics proteins of 49 kDa and 51 kDa (MiD49 and MiD51, respectively), and dynamin-related protein 1 (Drp1). (**B**) As senescence progresses, dysfunctional mitochondria are not efficiently eliminated and constitute a major cause of excessive ROS production. Increased oxidative damage by senescence depletes FIS1 and disrupts the mitochondrial fusion/fission balance, resulting in the formation of enlarged mitochondria. ROS: reactive oxygen species.

**Figure 2 cells-10-03003-f002:**
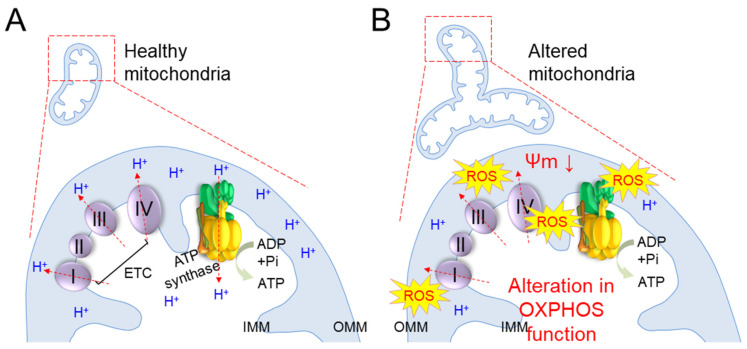
Schematic representation of basic mechanisms of senescence-induced mitochondrial metabolic changes. (**A**) Adenosine triphosphate (ATP) production is accomplished by a sequential reaction called oxidative phosphorylation (OXPHOS), which involves four electron transport complexes (ETC; complexes I through IV) and ATP synthase. During OXPHOS, the redox process of the ETC produces a hydrogen ion (H^+^) concentration gradient, leading to the movement of H^+^ from ATP synthase to the matrix to generate ATP. IMM: inner mitochondrial membrane; OMM: outer mitochondrial membrane. (**B**) During senescence, dysfunctional mitochondria produce excess reactive oxygen species (ROS), causing the unwanted oxidation of proteins involved in OXPHOS and impairment of their function. Thus, electron transport in the ETC is disturbed and electrons leak out of the ETC. Inefficient electron transport concurrently impairs proton transport through the IMM and dissipates ΔΨm, thereby reducing the efficiency of OXPHOS and accompanying a lack of ATP production. The leaked electrons also react with O_2_, generating excessive mitochondrial ROS.

**Figure 3 cells-10-03003-f003:**
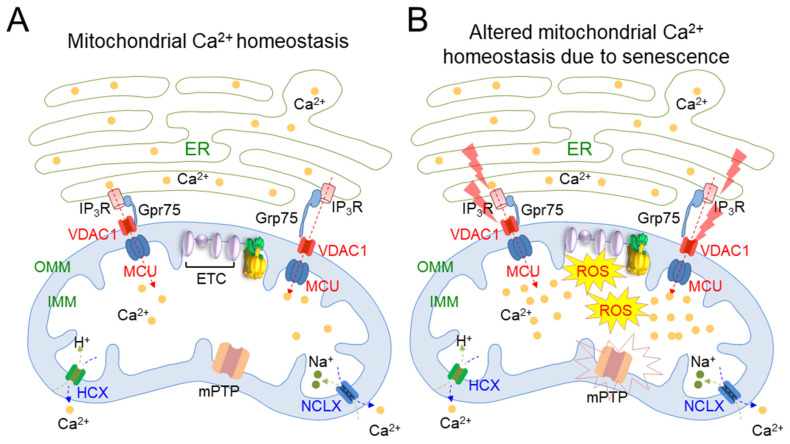
Schematic representation of the basic mechanisms of mitochondrial Ca^2+^ homeostasis. (**A**) Mitochondrial Ca^2+^ homeostasis is regulated by protein channels localized in the mitochondrial inner membrane (IMM) and mitochondrial outer membrane (OMM), and also by crosstalk with the ER. ER: endoplasmic reticulum; IP_3_R: inositol 1,4,5-trisphosphate receptor; VDAC: voltage-dependent anion channels; MCU: mitochondrial calcium uniporter; HCX: H^+^/Ca^2+^ exchanger; NCLX: Na^+^/Ca^2+^ exchanger; mPTP: mitochondrial transition pore. (**B**) Senescence triggers IP_3_R to release Ca^2+^ from the ER and causes VDAC/MCU channels to initiate the inward flow of Ca^2+^, leading to mitochondrial Ca^2+^ overload. Mitochondria overloaded with Ca^2+^ cause the collapse of the electron transfer in the ETC, resulting in increased electron leak and consequent mitochondrial ROS generation. An increase in mitochondrial ROS due to mitochondrial Ca^2+^ overload induces the sustained opening of the mitochondrial transition pore (mPTP). The lightning bolt indicates stress caused by senescence.

**Figure 4 cells-10-03003-f004:**
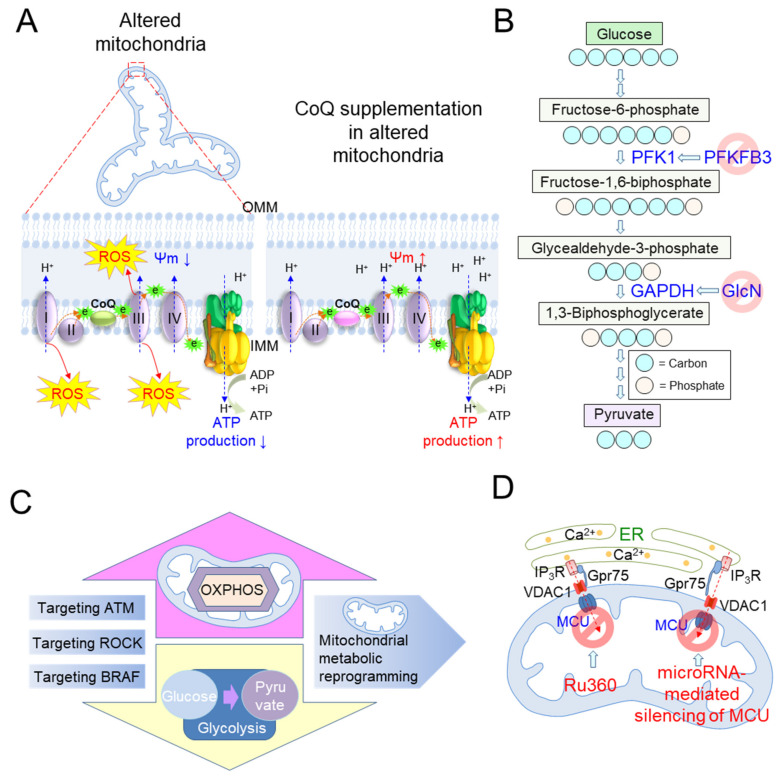
Targeting mitochondrial metabolism as a strategy to treat senescence. (**A**) Senescent cells exhibit a deficiency in coenzyme Q10 (CoQ), which accepts electrons from complex I/II and transfers them to complex III in the ETC (green CoQ indicates a deficiency in CoQ). Electrons that leaked from the ETC prematurely react with O_2_, causing excessive mitochondrial ROS production. A deficiency of CoQ reduces the efficiency of OXPHOS based on ΔΨm in flowing protons back to the mitochondrial matrix via ATP synthase, resulting in decreased ATP production. CoQ supplementation improves OXPHOS efficiency, which manifests as a significant increase in ATP production and a significant decrease in mitochondrial ROS generation (pink CoQ indicates a higher level of CoQ in the IMM). (**B**) *PFKFB3* is a gene that encodes 6-phosphofructo-2-kinase/fructose-2,6-biphosphatase 3, which activates phosphofructokinase 1 (PFK1), converting fructose-6-phosphate to fructose-1,6-bisphophate. Pharmacological inhibition of the glycolytic activator, PFKFB3, inhibits glycolysis and prevents the senescence-associated secretory phenotype (SASP)-mediated spread of senescence in endothelial cells (ECs). D-glucosamine (GlcN) inhibits the activity of glyceraldehyde-3-phosphate-dehydrogenase (GAPDH) in the glycolytic pathway. GlcN increases mitochondrial respiration by promoting the dependence of energy metabolism on OXPHOS while impairing glycolysis. (**C**) The strategy of targeting pathways directly or indirectly linked to mitochondrial metabolism. ATM: ataxia telangiectasia mutated; ROCK: rho-associated protein kinase; BRAF: serine/threonine protein kinase B-Raf. (**D**) The strategy of maintaining mitochondrial metabolism through mitochondrial Ca^2+^ homeostasis. The cell-permeable MCU inhibitor, Ruthenium 360 (Ru360), maintains mitochondrial Ca^2+^ at basal levels and improves OXPHOS efficiency. The inhibition of MCU by microRNA-mediated silencing also protects cardiomyocytes from oxidative damage and restores mitochondrial function restoration.

**Table 1 cells-10-03003-t001:** A summary of mitochondrial alterations associated with senescence.

Mitochondrial Alteration	Outcome(s)	Experimental Model and References
Alteration in mitochondrial morphology	Formation of giant mitochondria featuring highly interconnected networks	Human fibroblasts [20]
	A significant increase in the proportion of giant mitochondria	30-month-old C57/BL mice [22]
Alteration in mitochondrial function	Large aggregates of mitochondria with low ΔΨm and impaired ATP production	Senescent endothelial cells [24]
	The oxidative stress induced by rotenone and antimycin A deteriorates mitochondrial function	Human fibroblasts [25]
Alteration in OXPHOS function	Deterioration of the ETC complexes in liver, brain and muscle tissues Decrease in mitochondrial respiratory function	20-, 60-, or 100-week-old Wistar rat [26] Tissues from aged rats [27] 2-, 12-, 18-, or 24-month-old C57BL6 mice [28]
	A mouse model of senescence produced by *mev-1* (ortholog of the complex II) mutation exhibits deterioration of OXPHOS accompanying precocious age-dependent corneal physiological changes	Tet-mev-1 conditional transgenic mice [29]
	Decrease in complex II activity sustains the disruption of ΔΨm with significantly reduced intracellular ATP levels prior to the acquisition of the senescence phenotype	Chang cells [30] Hepatocyte cell lines [31]
	*TGF-β1*-mediated inhibition of complex IV directly triggers the senescence arrest in mink lung epithelial cells	Mink lung epithelial cells [32,33,34]
Decreasing dependence on OXPHOS but increasing dependence on glycolysis	Glycolysis is upregulated to generate additional ATP to compensate for the loss of energy production in dysfunctional mitochondria	Human coronary artery smooth muscle cells [35]
	The increase in glucose consumption and lactic acid production	Human fibroblasts [36]
	Significant transitions to more glycolytic states	Human fibroblasts [37]
Alteration in mitochondrial Ca^2+^ homeostasis	Senescence triggers IP_3_R to release Ca^2+^ from the ER and causes VDAC/MCU channels to initiate inward flow of Ca^2+^	Human endothelial cells and human fibroblasts [38]
	Mitochondria overloaded with Ca^2+^ causes the collapse of electron transport in the ETC	Human endothelial cells and human fibroblasts [38]
	Sustained opening of the mitochondrial transition pore (mPTP)	36-month-old C57BL/6J mice [39]
	Then, mPTP opening causes a rapid collapse in ΔΨm and swelling of mitochondria	Neural progenitor cells [40]

## Data Availability

Not applicable.

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
