# Peer review of "Targeting Mitochondrial Metabolism as a Strategy to Treat Senescence"

_cells, 2021, doi:10.3390/cells10113003_

Round 1
Reviewer 1 Report
In this review, Y H Lee et al. discuss the alteration in mitochondrial dynamics function associated to cellular senescence and the molecular pathways that could potentially block senescence processes. Several aspects have been discussed but some important additional information should be included in this work.
Mayor concerns:
- The authors indicate that the fission protein 1 (fis1) plays a relevant role in the mitochondrial fission process (line 69-71). However, the role of Fis1 in mitochondrial dynamics has been controversial during the last decade. The author should, therefore, include more recent references related to the mechanism by which Fis1 modulates mitochondrial dynamics.
- The interaction between DRP1 and MFF (mitochondrial fission factor) is critical to sustain mitochondrial fission. Please include a comment in the text and new references related to this point. In addition, the authors should include MFF in figure 1.
- In figure 3 the authors suggest that an interaction among Mfn2, ITPR2 and VDAC1 sustain the interaction between ER and mitochondria. Biochemical information related to the presence of this protein complex in the cell has been not reported. Actually, GRP75 mediates the interaction between ITPR2 (in the ER) and VDAC (in mitochondria). Please, explain and include references about this schematic representation or modify it in agreement with the recent data.
- The authors describe the role of PFKFB3 in senescence (line263-268). The references that the authors include to describe the role of PFKFB3 protein in senescence does not indicates it. Please review this paragraph.
- In order to show a summary of mitochondrial dysfunction associates to senescence, the authors can incorporate a “Summary Table” indicating the mitochondrial alterations present in senescence and the model in which they were found.
- In section 2, the authors could incorporate some reference/comment about the additional metabolic consequences of mitochondrial dysfunction during senescence (different to glycolysis); such as alterations in metabolomics profile or metabolic rewiring.
Minor concerns
The author should incorporate the original article reference in several paragraphs:
- Line 78-80: mitophagy paper
- Line 83 : role of Fis1 in senescence
- Line 107, 111: Role of mitochondria, p53 and pRB in oxidative stress and senescence
- The paragraph that describes OXPHOS in senescence says “exacerbation”, I believe that is a typo (line:134)
- I suggest the incorporation of the reference PMID: 26848154 , related with the role of mitochondria in senescence
Reviewer 2 Report
In this review Yun Haeng Lee and collaborators, focused their attention on mitochondrial dysfunction and alteration of mitochondrial metabolism that characterize cellular senescence.
Moreover they assess the significance of mitochondrial metabolic reprogramming for senescence regulation and propose strategies aimed to modulate mitochondrial metabolism to ameliorate senescence.
This review is very well written, extensively documented, well organized with high levels of graphical explanation.
Furthermore, I think that the authors have been very good at describing with a simple but at the same time technical and precise language, certain processes and details of mitochondrial metabolism, making them understandable even to those who are not in the sector.
Authors described changes occuring in bioenergetics, mitochondrial dynamics, membrane potential, Ca homeostasis and ROS production.
I think that the quality of manuscript maybe improved by considering the relevance of mitochondrial proteostasis and quality control in senescence (i.e.: doi.org/10.3389/fcell.2021.656201; doi.org/10.1016/j.cmet.2014.05.006).
Moreover, I think that the section “Targeting Mitochondrial Metabolism as a Strategy to Treat Senescence” maybe improved including other therapeutic strategies that modulate OXPHOS.
Here some example:
Caloric restriction (CR)/dietary restriction (DR) mimetics may represent one promising strategy for pharmacological targeting of senescent cells (doi:10.3390/nu12051344)
It has been demonstrated that resveratrol improves mitochondrial function and protects against senescence by inducing PGC-1a and SIRT1 activity (DOI: 10.1016/j.cell.2006.11.013); similar results were described using fibrates (doi.org/10.1016/j.ebiom.2019.07.018).
Moreover, it was recently reported the role for mitochondria in specific elimination of senescent cells using mitochondria-targeted tamoxifen (MitoTam), based on the capacity of non-proliferating non-cancerous cells to withstand oxidative insult induced by OXPHOS inhibition (doi.org/10.1038/s41418-018-0118-3).
Recently, it was reported that boosting mitochondrial function through a metabolic modulator called PD-0E7 resulted effective on counteract deleterious effects of precocious senescence in the SAMP8 mouse model (doi.org/10.3389/fphar.2020.01171).
Another emerging strategy aims to increase intracellular NAD+ levels with nicotinamide riboside (NR) to prevent senescence and SASP (doi.org/10.1111/acel.13329).
